# Impact of an Educational Intervention for Healthy Eating in Older Adults: A Quasi-Experimental Study

**DOI:** 10.3390/ijerph20196820

**Published:** 2023-09-25

**Authors:** José Ortiz Segarra, Ulises Freire Argudo, Dayanara Delgado López, Stalin Ortiz Mejía

**Affiliations:** Facultad de Ciencias Médicas, Universidad de Cuenca, Cuenca 010107, Ecuador; ulises.freire@ucuenca.edu.ec (U.F.A.); daya-delgado@hotmail.com (D.D.L.); jose.ortizm@ucuenca.edu.ec (S.O.M.)

**Keywords:** older adults, health education, healthy eating

## Abstract

The elderly population in Ecuador is increasing rapidly, with an increasing incidence of diet-related diseases. The elderly living in the community seek alternative and complementary methods to improve their diet and quality of life. This study aimed to investigate the impact of an educational intervention on knowledge related to healthy eating among older adults. This intervention is rooted in the principles of meaningful learning and incorporates culturally adapted materials. A quasi-experimental study design was employed using a pre-test–post-test control group. Study participants were a total of 109 elderly (intervention: *n* = 51, control: *n* = 58) people in Cuenca, Ecuador. The educational intervention based on Ausubel’s theory of significant learning and Vygotsky’s sociocultural theory was programmed to be carried out for one session per week, over 24 weeks, with a duration of 120 min per session. The measures were the general characteristics of the study participants and knowledge about necessary amounts, food sources and the consequences of deficits or excesses in the consumption of macro- and micronutrients. Data were collected from August 2018 to February 2019. Statistically significant differences were observed between the intervention group (IG) and the control group (CG) in terms of knowledge about healthy eating for older adults following the educational intervention. The outcomes of this study strongly suggest the efficacy of the program in improving knowledge related to healthy eating among older adults. Healthcare providers should prioritize food education based on meaningful learning, utilizing culturally adapted materials for the elderly individuals residing within the community.

## 1. Introduction

According to the World Health Organization (WHO), the percentage of older adults (OA) aged 60 and over is projected to increase by 34% between 2020 and 2030. By 2050, the number of older adults will exceed that of adolescents and youth ages 15–24, and approximately 65% of this population will live in low- and middle-income countries. In response to this demographic change, countries must prepare to ensure that their health and social systems can adequately address these changes [1].

In Latin America and the Caribbean, there has been a constant and continuous increase in life expectancy. From an average of approximately 59 years between 1965 and 1970, it has risen to almost 76 years in the period 2015 to 2020. The population has gained 17 years of life in the last 55 years [2]. In Ecuador, life expectancy was estimated at 75 years in 2010 and is projected to reach 80.5 years in 2050, which will translate into an increase in the elderly population. In 2020, out of a total population of 17,510,647, 1,904,568 were 60 years of age or older, representing 10.88% of the national population [3]. Among the determinants of longevity in individuals are genetic and environmental factors, as well as lifestyle, nutrition being a key component. An observational study conducted in the United Kingdom, which analyzed data collected between 2006 and 2010 with follow-up until 2016, involving 480,940 adults aged 38–73 years, concluded that a healthy lifestyle (including physical activity during leisure time, smoking, diet and alcohol consumption) is associated with an increase of up to 6.3 years in life expectancy for men and 7.6 years for women, even in the presence of chronic diseases [4].

In Ecuador, a high prevalence of malnutrition exists, evident both within the community (20.4%) [5] and in geriatric centers (82.61%) [6]. A study has reported that 72.2% of individuals are at risk of malnutrition, while 17.6% exhibit signs of malnutrition. In both cases, a significant association was identified with lower economic status (*p* = 0.001) and education level (*p* = 0.03) [7]. Another study conducted in Guayaquil, one of the most densely populated cities in the coastal region of the country, has highlighted that across all gerontology centers, 23% of the men were underweight and 16% were overweight and obese; for women, 26% were underweight and 29% were overweight or obese [8]. Additionally, research has indicated that among older adults with an average age of 76 years, who belong to clubs for the elderly linked to primary health care centers and have a low socioeconomic status, only 36% had adequate nutrition, 57% were nutritionally deficient, and 7% exhibited excessive nutritional intake [9]. Therefore, the necessity of conducting educational interventions in nutrition for the elderly population is evident, while also considering the critical role of social and economic determinants.

Several studies have shown that adequate nutrition as part of a healthy lifestyle promotes the maintenance of cognitive abilities and stimulates the immune system [5,6,7,8,10,11,12,13]. Diets low in carbohydrates or rich in vegetables, fruits, nuts, whole grains, fish and unsaturated fats that contain antioxidants, potassium and omega-3 reduce the risk of cardiovascular disease and obesity and protect the brain, promoting a healthy life [14]. Conversely, high consumption of (red) meat, especially processed meat, is associated with higher all-cause mortality. Furthermore, the Mediterranean diet and other high-quality diets are associated with a reduced risk of all-cause mortality [15,16]. One of the mechanisms through which diet can reduce disease risk is via its impact on telomere length. Telomeres are the ends of chromosomes that play a crucial role in their structural stability, cell division and lifespan. Telomere length is highly correlated with chronological age and metabolic status, and people with shorter telomeres are at higher risk of chronic disease and mortality. Diet can influence telomere length through several mechanisms, such as the regulation of oxidative stress and inflammation or the modulation of epigenetic reactions [17,18].

A systematic review and meta-analysis show that healthy diets are associated with a reduced risk of all-cause mortality, the incidence of type 2 diabetes, cardiovascular disease, and cancer in general, particularly colorectal cancer [19]. Although the biological changes that occur through the natural aging process cannot be controlled, lifestyle-related risk factors can be modified through education. A strong level of knowledge, along with favorable attitudes related to nutrition, can exert a positive influence on the health and overall quality of life within the elderly population [20]. This is due to the pivotal role that optimal nutrition plays in preventing a multitude of age-related diseases and in enhancing the quality of life of the elderly [21]. Notably, a higher educational level has been associated with heightened nutritional awareness in older individuals [22,23]. Proficiency in knowledge, coupled with positive attitudes and adept nutritional practices, holds paramount importance in the battle against malnutrition and the promotion of robust health [24], hence the importance of this study.

According to scientific evidence, a healthy diet would be one in which fats do not exceed 30% of the total caloric intake, to avoid weight gain [25,26]. This implies a reduction in saturated fat consumption and their substitution with unsaturated fats [26], the gradual elimination of industrially produced trans fats, and the limitation of free sugars intake to less than 10% of the total caloric intake [27,28]. Keeping salt intake below 5 g per day helps prevent hypertension and reduces the risk of heart disease and stroke in the adult population [29]. WHO member states have agreed to reduce salt intake among the world population by 30% and halt the rise in adult and adolescent obesity and diabetes, as well as childhood overweight, by 2025 [27].

Older adults face malnutrition risks due to physiological, psychological, environmental, social and dietary factors [29]. In response to this challenge, research has been conducted employing effective didactic strategies intended to foster positive influences on appropriate dietary habits and lifestyles, ultimately contributing to healthy aging [30]. In this context, certain studies have demonstrated a correlation between nutritional education and shifts in health behaviors across diverse population segments [30,31]. Additionally, publications have drawn the conclusion that patients and family group education forms the foundation for managing non-communicable diseases such as diabetes [32,33,34,35].

In the scientific literature, several authors coincide in identifying at least nine key factors that influence food choice, such as knowledge of the product, the sensory appeal of the food, the convenience of purchase and preparation, the perception of the food as “natural”, the consideration of food as healthy or unhealthy, weight control, influence or regulation of mood and ethical considerations regarding food production and place of origin. In addition, learning, experience and exposure to food play an essential role [5,6,7,8]. In general terms, health programs that include educational interventions have shown significant effects on quality of life, both in aspects of mental health and physical health [36,37] and in clinical conditions of individuals, such as systolic blood pressure [38], diastolic blood pressure [39], cholesterol levels, urinary sodium, abdominal circumference [40] and body mass index (BMI) [41].

The nutritional status of older adults is contingent not only upon their familiarity with healthy eating practices [42,43], but also on various social, environmental and health determinants pertinent to the elderly [44]. Among the social factors intertwined with nutrition are lifestyle, loneliness, isolation, marital status, educational level, socioeconomic status and place of residence [45]. In this regard, the dissemination of knowledge becomes imperative for effectively enhancing dietary patterns and overall health. A meta-analysis [46] and a systematic review [47] have underscored noteworthy and positive correlations between knowledge, attitudes and dietary intake.

## 2. Materials and Methods

### 2.1. Participants

The target population comprised 109 individuals aged 65 years and above, residing in the areas of influence of four primary health care centers within rural communities belonging to the city of Cuenca, Ecuador. The communities were Victoria de Portete and Cumbe, which made up the control group, and Checa and Chiquintad, which encompassed the intervention group. Situated to the south, Victoria de Portete and Cumbe are positioned at distances of 23 km and 28 km from Cuenca, respectively. Victoria de Portete has a population of 5251 individuals, including 2391 men and 2860 women; Cumbe, on the other hand, is home to 5546 people, of which 2480 are men and 3066 women. The communities of Checa, with a populace of 4826, comprising 2251 men and 2575 women, and Chiquintad, accommodating 2741 inhabitants, featuring 1182 men and 1559 women, are in close proximity to each other, situated 13 km northeast of the city of Cuenca.

In terms of sociodemographic characteristics, the populations of these four parishes exhibit similarities. Their primary income sources revolve around diverse paid endeavors such as construction, commerce, and manufacturing industries, among others. Both men and women engage in domestic employment. Additionally, during their leisure time, all members of the families participate in agricultural and livestock activities, as well as engaging in the production of textiles and handicrafts as a means of self-subsistence.

### 2.2. Study Design

A quasi-experimental intervention study was conducted using convenience and allocation sampling. The participants from the communities of Victoria de Portete and Cumbe were assigned to a control group (CG) (*n* = 58) and the participants from the communities of Ochoa León and Uncovía to the intervention group (IG) (*n* = 51), using a prospective pre–post design. Between the communities of the intervention group and those of the control group there is a distance of 30 km, through which mutual influence was avoided.

The sample was determined using the Epi Info program version number 7.2.3.1 (Center for Disease Control and Prevention, Atlanta, United States), employing the following criteria:
-Two-sided confidence level: 95%;-Power: 80%;-Ratio (unexposed: exposed): 1;-Percentage of outcome in unexposed group: 20%;-Risk ratio: 2;-Odds ratio: 3;-Percentage of outcome in exposed group: 45%;-Sample size: 56 for each group.

### 2.3. Intervention

The educational intervention was based on Ausubel’s theory of meaningful learning [48] and Vygotsky’s Sociocultural Theory [49]. Meaningful learning occurs when novel information becomes linked with pertinent concepts that are already embedded within the cognitive structure of the students. This entails that fresh ideas, concepts and propositions can be absorbed in a meaningful manner, provided that they are lucid and accessible, serving as pivotal reference points for initial comprehension. Concurrently, this novel knowledge modifies the cognitive frameworks, amplifying the cognitive schemas conducive to assimilating further knowledge. This process hinges upon two crucial requisites: firstly, the educational material must possess inherent potential for meaningful engagement by the learner; secondly, a motivational inclination or willingness to learn must be present. Significant learning constitutes a comprehensive process encompassing the emotional, motivational and cognitive dimensions of the individuals [50,51]. In this study, solely the cognitive dimension was assessed, while the remaining two dimensions were incorporated into the didactic strategy. These dimensions are elaborated upon both within the development process of the educational intervention and within the culturally adapted materials generated. This theory is further augmented by the perspectives of Vigotsky, which posit that within the learning journey, cultural development initially evolves as a social and interpsychological process, subsequently transforming into an individual and intrapsychological experience. The trajectory of learning extends from the external to the internal sphere through interactions and relationships with fellow individuals within the community [52,53].

On this basis, the importance of the previous cognitive structure of older adults was evaluated, considering structured teaching as the core of the learning process. The educational objective was to promote learning about the value of healthy eating. The educational process consisted of five stages: (1) motivation, to achieve the adherence of the participants to the proposal, through local popular games; (2) problematization and reflection, for the development of content with cultural adaptation; (3) coping, to obtain the necessary local family and community resources for educational activities and the reinforcement of knowledge and practices; (4) resolution, with the purpose of establishing personal and collective commitments for the development of healthy eating practices; and (5) evaluation, to verify the achievement of educational objectives. The whole process aimed to improve knowledge (knowing) and the ability to internalize (knowing how to learn), laying the foundations for the future development of technical and procedural skills (knowing how to do), as well as attitudes (knowing how to be) and social skills (knowing how to live together) in the field of nutrition for the well-being of the elderly.

Drawing upon a comprehensive review of the scientific literature and meticulous analysis of the knowledge of the elderly population, both the educators and senior medical students collaboratively assembled an illustrative brochure comprising 19 informative responses pertinent to the questions featured within the knowledge assessment forms. The brochure encompassed the following topics: (1) daily protein requirements, (2) sources of protein-rich foods, (3) implications of inadequate protein intake, (4) daily carbohydrate requirements, (5) carbohydrate-rich food sources, (6) consequences of excessive carbohydrate consumption, (7) daily fat prerequisites, (8) food items containing fats, (9) consequences of excessive fat consumption, (10) impacts of low fat intake, (11) daily vitamin and minerals necessities, (12) foods enriched with vitamins and minerals, (13) consequences of deficient vitamins and mineral intake, (14) daily water requirements, (15) water-containing foods, (16) consequences of insufficient water intake, (17) daily fiber demands, (18) fiber-rich dietary sources and (19) consequences of an inadequate fiber intake.

Subsequently, a questionnaire comprising the 19 assessment queries was tailored in accordance with the contents of the brochure. The evaluation responses were graded utilizing a Likert scale ranging from 0 to 4 for each question. A score of 4 was attributed when the participant correctly addressed 4 of the concepts elucidated in the brochure; 3 points were granted for accurately explaining 3 of the concepts, followed by 2 points for correct responses on 2 items, and 1 point for the accurate addressing of a single item. Conversely, a score of 0 was assigned for incorrect or omitted responses.

The initial evaluation was conducted for both the intervention and control groups, prior to the commencement of the educational program. Subsequently, the second evaluation was administered six months after the initiation of the intervention, concurrently for both groups.

Educational resources used to deliver the content included traditional folk games for older adults, illustrated booklets, vignettes or illustrated stories, videos, plastic figures and posters illustrating food sources, macronutrients, micronutrients, water and fiber. Both the content and the educational resources were culturally adapted. The program was implemented in community houses and health center auditoriums by students in the community rotation phase of the internship program. Prior training was provided and students were continuously monitored by teachers for a period of 6 months. Each session had an average duration of 2 h, once a week, totaling 48 h. All participants who entered the study remained until the end, so there were no losses to follow up. To facilitate an accurate depiction of food consumption, participants were provided with standardized containers for daily use, and the research team quantified the quantities in milliliters (mL) and grams (g) for both liquid and solid food items. For instance, a small spoon equated to 5 mL or g, while a large spoon represented 10 mL or g. Similarly, a small teacup or cup denoted 150 mL or g, and a larger coffee cup indicated 250 mL or g. Additionally, a flat plate was equivalent to 150 mL or g, while a deep plate, such as a soup bowl, corresponded to 250 mL or g.

### 2.4. Data Collection

Prior to the application of the knowledge and good eating practices, the reliability of the internal consistency of the scale was evaluated for the questions related to the educational intervention in other health centers with similar characteristics, resulting in a Cronbach’s Alpha coefficient of 0.76. Eligible participants who gave their consent were interviewed by the same students who carried out the educational intervention after going through a training process. The purpose of the form was to collect data in several segments. In the initial part, it included sociodemographic information such as gender, marital status, educational level and who they live with; and in the second part, questions related to educational content were included. This form was administered before the educational intervention and six months later, at the end of the process. Subsequently, a questionnaire comprising the 19 assessment queries was tailored in accordance with the contents of the brochure (See Table A1, Appendix B). 

### 2.5. Statistical Analysis

Statistical analyses were conducted utilizing SPSS version 20.0 for Windows. Continuous data were presented as mean and standard deviation (SD), while categorical data were expressed as numbers (*n*) and percentages (%). Data distribution normality was assessed via the Kolmogorov–Smirnov test, yielding a *p*-value > 0.05 for both the CG and the IG. Disparities in knowledge regarding healthy eating before and after the intervention were assessed using the Odds Ratio (OR) with its corresponding 95% confidence interval (CI) and the Chi-square test. A composite indicator, derived from statistically weighting participant responses through Principal Component Analysis, was employed. Associations between demographic characteristics and the outcome variable, quantified using the composite indicator, were investigated using bivariate logistic regression.

### 2.6. Ethical Considerations

This study was carried out in accordance with the ethical principles of the Declaration of Helsinki and received the approval of the Bioethics Committee of the University of Cuenca under code 2018-007EO. The participants were voluntarily recruited. The interviewers explained the objectives and procedures of the study to the subjects. Signed informed consent or a fingerprint (for illiterate participants) was obtained prior to inclusion. The confidentiality of the data collected was ensured and access to the information was limited to the study team. All volunteers, regardless of their inclusion in the study, were welcome to participate in the educational meetings.

## 3. Results

A total of 109 older adults of both genders participated in the current study. Figure 1 offers demographic details and baseline variables concerning the participants. Prior to the intervention, no statistically significant differences in demographic characteristics between the groups were observed, encompassing factors such as gender, marital status, educational level, cohabitation arrangement and economic stratum, with the exception of age (Appendix A).

Figure 2 presents the bivariate analysis of the differences in the means of the composite indicator between the CG and the IG, before and 6 months after the educational intervention, as well as the possible association between said differences and the demographic variables. Before the educational intervention, no statistically significant differences were observed in the mean score of the composite indicator of knowledge between the groups (OR 1.35; 95% CI 0.63–2.90; *p* > 0.05), while after the educational intervention, statistically significant differences were found (OR 3.74; 95% CI 1.43–9.79; *p* < 0.05). Prior to the educational intervention, none of the demographic variables were associated in a statistically significant way, except for the age group of 65 to 74 years (OR 5.30; 95% CI 1.76–15.95; *p* < 0.05). In the post-intervention phase, three variables associated in a statistically significant way with the difference in the mean score of the composite indicator of knowledge between the groups were identified; namely, the female gender (OR 5.66; 95% CI 2.2–14.59; *p* < 0.001), living with other relatives (OR 4.01; 95% CI 1.31–12.24; *p* < 0.05) and the middle economic stratum (OR 6.13; 95% CI 2.37–15.85; *p* < 0.001) (Appendix A).

Figure 3 illustrates that none of the demographic characteristics were associated with the differences in the averages of the composite indicator related to knowledge about healthy eating between the CG and the IG after the educational intervention, except the age group from 75 to 84 years, taking as reference the group of 65 to 74 years. For each of the demographic characteristics, a reference group was also considered, namely, for the gender variable, male; for the marital status, married; for educational level, primary; for who do you live with, alone; for economic level, high. Although the age group of 75 to 84 years influenced the knowledge differences between the groups after the educational intervention, it is necessary to consider that this difference depends on the other demographic variables that do not affect the results of the educational intervention (Appendix A).

## 4. Discussion

The favorable outcomes, manifested in the enhanced comprehension of healthy eating among elderly individuals from rural communities, as achieved through an educational intervention based on the theory of Ausubel’s of meaningful learning [48] and the sociocultural theory of Vygotsky [49], were made possible due to several key factors. These encompassed the establishment of a meaningful and substantial connection between the learning materials and the previous knowledge of the participants, the provision of logically structured, culturally adapted learning resources, and the effective engagement of participants through popular games.

During the initial bivariate analysis examining differences in knowledge between the groups, it became apparent that factors such as gender, cohabitation arrangements, and economic status could potentially influence the observed increase in knowledge within the intervention group. However, subsequent logistic regression analysis revealed that only age remained a contributing factor, contingent upon the presence of the other demographic variables considered within this study.

Our findings suggest that an educational intervention based on significant learning and sociocultural theory, which includes popular games with culturally adapted content for older adults, may be effective in improving knowledge about healthy eating. There were statistically significant increases in the average number of correct responses on the post-test in the IG, but not in the CG. However, previous studies on nutrition-focused educational interventions in older adults based on meaningful learning have not been reported. Our results are comparable with findings from other research using various methodologies, which have shown significant improvements in nutrition knowledge, nutritional status and quality of life [54,55,56,57,58]. For example, the Edumay project carried out in the municipality of Pamplona demonstrated statistically significant differences in the adherence to the Mediterranean diet of the intervention group three months after participating in the educational program, while the control group showed no variation [55].

The Educational Program for the Self-Care of the Elderly (PECA), which focuses on the quality of life, nutritional status and perceived social support of older adults who live in their own homes, showed a statistically significant difference in the scores of the mini nutritional assessment in the intervention group during the post-intervention, but not in the control group [56]. In Mexico, a quasi-experimental study was conducted that compared pre- and post-intervention with a comparison group of 22 hypertensive adult patients without comorbidities seen in primary care centers. The study group (*n* = 11) received group education and consultations on nutrition, while the control group (*n* = 11) received only nutrition consultations. The study achieved a reduction in weight and body mass index and increased knowledge [58]. An investigation carried out in Arequipa, Peru, demonstrated an improvement in the lifestyle of older adults who participated in health circles, particularly in the dimensions of family, self-reflection and nutrition/food. Differences in final scores as well as in pre- and post-intervention scores by dimensions were statistically significant [57]. While our study solely focused on assessing healthy eating knowledge within older adult community organizations, similar results were obtained post-intervention.

In the present study, it was found that older adults had pre-existing knowledge about healthy eating, with a percentage ranging from 6 to 14%, which is comparable to other studies [37,54,55,57,58]. Considering that healthy aging is a continuous process of the optimization of opportunities to maintain and improve physical and mental health, independence and quality of life throughout life [59], a healthy diet is necessary to provide the nutrients that the body needs for proper functioning and to maintain or recover health, minimize the risk of disease and ensure a good quality of life.

This study was subject to several limitations stemming from constraints in material resources and time, which precluded the longitudinal tracking of participants to assess the influence of the intervention on healthy eating practices and their subsequent health impacts. Such an evaluation could involve measurements of biological indicators, including anthropometric measurements and laboratory tests. Additionally, this study was conducted with a relatively small sample size, which restricts the ability to demonstrate a statically significant change resulting from the implemented intervention.

Future research should focus on health care providers to assess their knowledge, attitudes, practices and training regarding healthy eating for older adults. Officials of the Ministry of Health must consider appropriate information and communication technologies for older adults. Health officials should also develop recommendations to improve the capacity of health workers to promote healthy eating education for older adults. Studies have shown that health care provider recommendations can positively influence the acceptance of healthy eating among older adults [38,55,59,60].

There is a need for a comprehensive strategy to educate older adults about healthy eating. Since doctors and nurses are the primary source of knowledge about healthy eating for older adults. providers can play a crucial role in encouraging older adults to adopt health-protective behaviors, dispel misconceptions about the evidence of screening for malnutrition and provide comprehensive health education to older adults to keep them committed to their health.

## 5. Conclusions

In this study, we developed and evaluated an educational proposal based on Ausubel’s significant learning and Vygotsky’s sociocultural theory to verify its effectiveness in improving knowledge about healthy eating in the elderly. The results in this study showed knowledge about necessary daily consumption, the foods that contain necessary nutrients and the consequences for the health of the deficits or excesses in the consumption of nutrients such as proteins, carbohydrates, fats, vitamins, minerals, water and fiber. This improvement can be attributed to the 48 h of effective intervention performed over a 6-month period, with an average of 2 h per session. A sequential process was developed and motivation was provided through culturally adapted content and popular games, allowing the benefits of good nutrition and the consequences of poor nutrition to be compared. Subsequently, educational activities based on appropriate prior knowledge were carried out using appropriate didactic materials for older adults. The findings demonstrated the importance of developing end-of-life care programs based on various teaching and learning strategies.

## Figures and Tables

**Figure 1 ijerph-20-06820-f001:**
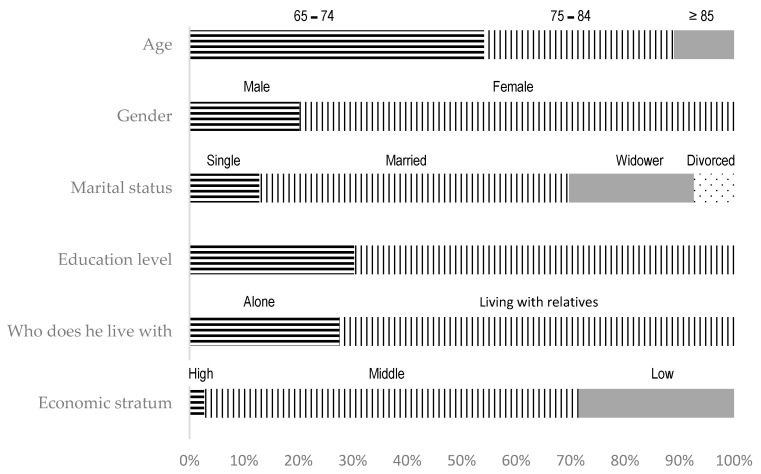
Demographic characteristics of the participants.

**Figure 2 ijerph-20-06820-f002:**
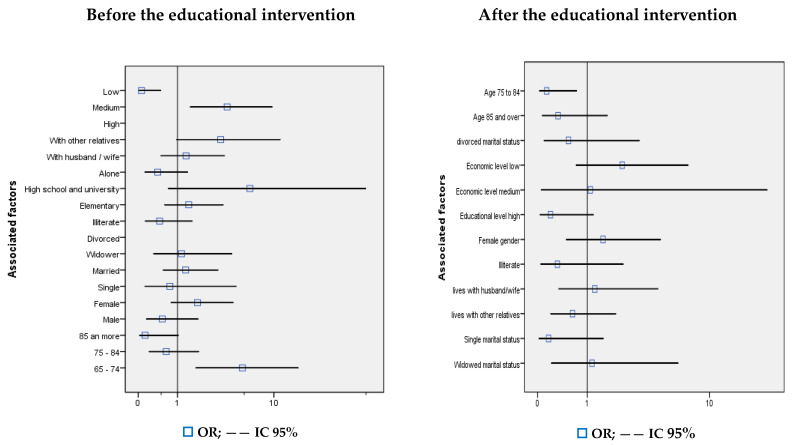
Bivariate analysis of the difference in knowledge before and after the educational intervention and associated demographic characteristics.

**Figure 3 ijerph-20-06820-f003:**
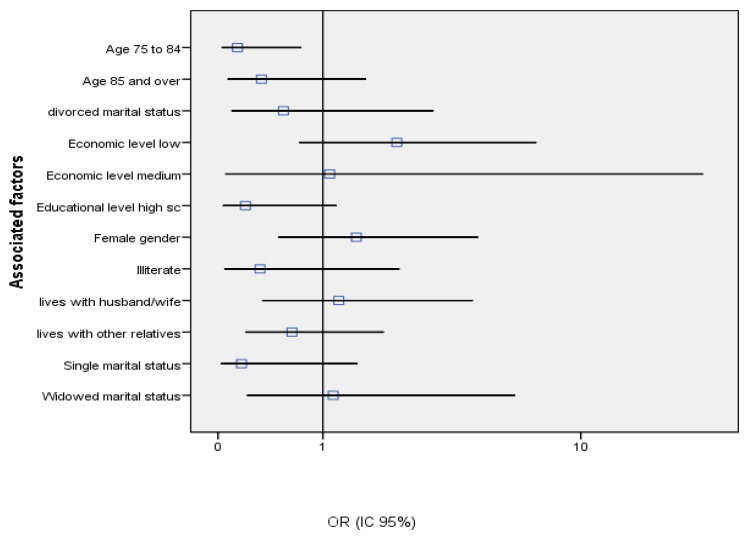
Logistic regression of demographic characteristics associated with knowledge about healthy eating, after the educational intervention.

## Data Availability

The data presented in this study are available on request from the corresponding author. The data are not publicly available due to privacy of information.

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
