# Peer review of "Impact of an Educational Intervention for Healthy Eating in Older Adults: A Quasi-Experimental Study"

_ijerph, 2023, doi:10.3390/ijerph20196820_

Round 1
Reviewer 1 Report
Dear Author
The table one not depict the economic criteria which is most important with food quality and amount.
The table 2 and table 3 describe about different Protein, Carbohydrate, water content, vitamin - mineral etc of food. But how the respondent can properly describe the amount in their consumed food is not clear.
Kindly clarify properly.
Advised to add graphical representation of the tables.
Author Response
Answer 1. The economic criteria were included in table one.
Answer 2. the following paragraph was added to the document.
"To facilitate accurate depiction of food consumption, participants were provided with standardized containers for the daily use, and the research team quantified the quantities in milliliters (ml) and grams (g) for both liquid and solid food items. For instance, a small spoon equated to 5 ml or g, while a large spoon represented 10 ml or g. Similarly, a small tea cup or cup denoted 150 ml or g, and a larger coffee cup indicated 250 ml or g. Additionally, a flat plate was equivalent to 150 ml or g, while a deep plate, such as a soup bowl, corresponded to 250 ml or g".
Answer 3. Due to the large volume of data described in the tables, a graphical representation was not possible.
Reviewer 2 Report
Thanks to authors for the interesting study and attempting to develop a culturally appropriate intervention. Please see my feedback on different sections noted below, which can help improve the content further.
Abstract:
Line 10-11 mentioned assessing 'the effects of an 10 educational intervention on healthy eating', but this study only assessed change in knowledge which is not practice. This needs revision.
Line 19-20: Need rewriting as it seems incomplete ("The differences in knowledge means between the experimental group (GE) and the control group (CG) after the educational intervention were statistically significant). The authors probably meant that there was a statistically significant between group difference in knowledge after the educational intervention.
Line 20-21: As claimed, this study do not suggest that the program is effective in improving both knowledge and practices as practices were not assessed, only change in knowledge was assessed. Please revise the sentence.
Introduction:
The introduction focuses a lot on healthy lifestyle and then briefly mentioned that knowledge promotion can be a means of healthy lifestyle (line 65-66), which should be backed with references. Improving diet and knowledge about diet are not the same. Previous references were in relation to effect of practice.
The author later provided references about knowledge promotion and link to health, which (references) should come in the introduction. The authors also need to talk a bit more about what it takes for knowledge promotion to successfully improve diet/health (e.g., socioeconomic aspects could be a barrier).
Line 70-71: "Limit the intake of free sugars to less than 10% of the total caloric intake" - This seems to be an incomplete sentence. It does not look like a new sentence but seems to be connected to the previous line so should be a continuation, joined by a comma to say - ", and limiting the intake of free sugars to less than 10% of the total caloric intake" .
Line 77-78: Again this seems to be another incomplete sentence, which need revision. The sentence ("Hence the importance of improving their knowledge about healthy eating, as well as....") is not expected to start with hence as such on its own without connecting to some other logical impact.
Methods
Study design: Line 109-111: Provide more information on how are these two communities similar or different so even though convenient sampling is done readers will have more contextual information available.
Intervention: Line 116-117- Add few lines highlighting the key focus/mechanisms of the Ausubel's theory and Vygotsky's theory so that readers who are not familiar with them have some broad overview.
2.5 statistical analysis: this section is very brief, mentioned normality tests but then did not say if the variables were found to be normally distributed or not. Line 162-164: for the paired t-tests, normality assumption need to be satisfied, but no information is provided.
Ideally, the authors needed to do regression analysis to examine effect of intervention on knowledge, adjusted for baseline knowledge status (as there is a consistent difference between group- the control group seems to have started with low knowledge for almost all variables included in the table 2.
Results:
All Tables should provide results displayed as decimal points separated by decimal points and not by comma (e.g., Table 1 Line 188-189, P value 0,1152 should be written as 0.1152).
Table 1 also did not include any socioeconomic information apart from educational level. more information about the groups should be provided, if collected.
All results in table 1 and elsewhere presenting means should be accompanied by Standard deviation (SD). The results for p-values which are shown as 0.000 should be reported as <0.001. The outputs are showing it as 0.000 as the exact value has lots of zeros before the number but the value is actually <0.001.
The labels used in Table 1, such as '1. Protein amount and reasons' are not good description of the question asked. The questions asked as it is should be included as an appendix.
Lie 213: Table 3 results examined individual item response and average for all item summed. Ideally, a composite indicator should be created by statistically weighting the items using the Principal Component Analysis technique. If not possible, mention this as a limitation as currently while averaging of all items, each item is considered equally important but this may be not true.
Provide a footnote for '17. Daily fiber intake and reasons' for Table 2 why this data is 0.00 for control group.
Table 3: Analysis should look into group difference adjusted for pretest status, in a regression model, from average of all the questions, rather than using paired t-test. There is a systematic bias for control group as they seems to have lower scores at baseline across all indicators, which is surprising. Please explain in discussion why that may be the case.
The ideal analysis would be to use the composite score variable as a single outcome, which is calculated by applying Principal Component Anlaysis (PCA- data reduction approach) on the questionnaire items, and then use that outcome in a regression model that also include baseline scores as explanatory variable. Since the baseline scores are not similar for the two groups, the results for group difference should be adjusted for baseline scores. If the authors can not use the PCA technique but prefers to use the summed overall score (considering equal weight for each questionnaire items) it should be mentioned in discussion/limitation.
Discussion:
Line 245-246: The study used very small sample for showing significant change, please also mention that.
Line 262-263: For limitation, it is not that this study did not examine long term impact, the current analysis is also limited as it only analysed change in knowledge rather than in practice (on health/nutrition). Each setting where a study is conducted is unique, so it can not be guaranteed if knowledge shown to improve behaviour in one setting can be similarly translated to positive effects in another setting. This study also did not account for socioeconomic status of the population, which often limits what choices the participants can make.
The authors have generally written good English as a whole, but there are few loose sentences hanging in places (some are included in my comments). Careful proof reading and editing with proper punctuation can resolve this.
Author Response
Answer 1. The following paragraph was included in the document.
“This study aimed to investigate the impact of an educational intervention on the knowledge re-lated to healthy eating among older adults”.
Answer 2. The following paragraph was included in the document.
“Statistically significant differences were observed between the intervention group (IG) and the control group (CG) in terms of knowledge about healthy eating for older adults following the educational intervention”.
Answer 3. The following paragraph was included in the document.
“The outcomes of this study strongly suggest the efficacy of the program in improving knowledge related to healthy eating among older adults”.
Answer 4. The following paragraph was included in the document.
“A strong level of knowledge along with favorable attitudes related to nutrition can exert a positive influence on the health and overall quality of life within the elderly popula-tion (20). This is due to the pivotal role that optimal nutrition plays in preventing a multitude of age-related diseases and in enhancing the quality of life of the elderly (21). Notably, a higher educational level has been associated with heightened nutritional awareness in older individuals (22,23). Proficiency in knowledge, couple with positive attitudes and adept nutritional practices, holds paramount importance in the battle against malnutrition and the promotion of robust health (24)”.
Answer 5. In the introduction of the corrected article, references were included on the promotion of knowledge and the link with health, as well as on what is needed for the promotion of knowledge to successfully improve diet and health.
Answer 6. The writing errors had been fixed in the corrected document.
Answer 7. The writing errors had been fixed in the corrected document.
Answer 8. The following paragraph was included in the document.
“The target population comprised 109 individuals aged 65 years and above, residing in the areas of influence of four primary health care centers within rural communities be-longing to the city of Cuenca, Ecuador. The communities were: Victoria de Portete and Cumbe, who participated as the control group, and Checa and Chiquintad, en-compassed within the intervention group. Situated to the south, Victoria de Portete and Cumbe are positioned at the distances of 23 km and 28 km respectively. Victoria de Portete has a population of 5,251 individuals, including 2,391 men and 2,860 women; Cumbe, on the other hand, is home to 5,546 people, of which 2,480 are men and 3,066 women. The communities of Checa, with a populace of 4,826, comprising 2,251 men and 2,575 women, and Chiquintad, accommodating 2,741 inhabitants, featuring 1,182 men and 1,559 women, are in close proximity to each other, situated 13 km northeast of the city of Cuenca.
In terms of sociodemographic characteristics, the populations of these four par-ishes exhibit similarities. Their primary income sources revolve around diverse paid endeavors such as construction, commerce, and manufacturing industries, among oth-ers. Both men and women engage in domestic employment. Additionally, during their leisure time, all members of the families participate in agricultural and livestock activi-ties, as well as engage in the production of textiles and handicrafts as a means of self-subsistence.
Answer 9. The following paragraph was included in the document.
“Meaningful learning occurs when novel information becomes linked with pertinent concepts that are already embedded within the cognitive structure of the students. This entails that fresh ideas, concepts, and propositions can be imbibed in a meaningful manner provided that they are lucid and accessible, serving as pivotal reference points for initial comprehension. Concurrently, this novel knowledge modifies the cognitive frameworks, amplifying the cognitive schemas conducive to assimilating further knowledge. This process hinges upon two crucial requisites: firstly, the educational material must possess inherent potential for meaningful engagement by the learner; secondly, a motivational inclination or willingness to learn must be present. Significant learning constitutes a comprehensive process encompassing the emotional, motiva-tional, and cognitive dimensions of the individuals (50,51). In this study, solely the cognitive dimension was assessed, while the remaining two dimensions were incor-porated into the didactic strategy. These dimensions are elaborated upon both within the development process of the educational intervention and within the culturally adapted materials generated. This theory is further augmented by the perspectives of Vigotsky, which posit that within the learning journey, cultural development initially evolves as a social and interpsychological process, subsequently transforming into an individual and intrapsychological experience. The trajectory of learning extends from the external to the internal sphere through interactions and relationships with fellow individuals within the community (52,53)”.
Answer 10. The following paragraph was included in the document.
“Data distribution normality was assessed via the Kolmogorov-Smirnov test, yielding a p-value > 0.05 for both the CG and the IG”.
Comment: The value of the normality test has been included in the statistical analysis, in the corrected document.
Answer 11. Regression analysis has been performed to examine the effect of the intervention on knowledge, adjusted for baseline knowledge status.
Comment: Included in the corrected document.
Answer 12. Commas have been replaced by points in the decimal numbers of the tables
Comment: Included in the corrected document.
Answer 13. In all the tables the economic level variable is analyzed.
Comment: Included in the corrected document.
Answer 14. Results in Table 1 included the standard deviation (SD) and results for p values shown as 0.000 have been reported as < 0.05 and < 0.001.
Answer 15.
Comment: The text of the questions on the form are described in an appendix.
Answer 16.
The following paragraph was included in the document.
“A composite indicator, derived from statistically weighting participant responses through Principal Component Analysis, was employed. Associations between demo-graphic characteristics and the outcome variable, quantified using the composite indi-cator, were investigated using bivariate logistic regression”.
Answer 17. An explanation is not needed because in the new analysis the score of each question is not compared, but rather the average of a composite indicator that was obtained by statistically weighting the responses of the participants using the Principal Component Analysis technique.
Answer 18. Considering that the control group presented lower scores at the beginning in all the indicators, a unique result was obtained, from the analysis of principal components (PCA data reduction approach) of the initial state of the test; This result was then used in a regression model, which included reference scores, in order to show whether or not the difference in knowledge is associated with the demographic variables.
Answer 19.
The following paragraph was included in the document.
“Disparities in knowledge regarding healthy eating before and after the intervention were assessed using the Odds Ratio (OR) with its corresponding 95% confidence in-terval (CI) and the Chi-square test. A composite indicator, derived from statistically weighting participant responses through Principal Component Analysis, was em-ployed. Associations between demographic characteristics and the outcome variable, quantified using the composite indicator, were investigated using bivariate logistic re-gression”.
Answer 20.
The following paragraph was included in the document.
“Additionally, this study was conducted with a relatively small sample size, which re-stricts the ability to demonstrate a statically significant change resulting from the im-plemented intervention”.
Answer 21.
The following paragraph was included in the document.
“This study was subject to several limitations stemming from constraints in material resources and time, which precluded the longitudinal tracking of participants to assess the influence of the intervention on healthy eating practices and their subsequent health impacts. Such evaluation could involve measurements of biological indicators, includ-ing anthropometric measurements and laboratory tests. Additionally, this study was conducted with a relatively small sample size, which restricts the ability to demonstrate a statically significant change resulting from the implemented intervention”.
Answer 22. Fixed single sentences and punctuation marks.
Reviewer 3 Report
The authors tried to show the effect of education intervention on nutritional knowledge of the elder people in Ecuador. But they did not measure the effect of intervention on practice of them. Below you can find some of my comments:
Line 29, abbreviation of older adults should be corrected to OA.
In introduction, it is necessary to mention the situation of eating in elder person in Ecuador and show the importance of conducting this intervention there.
How did you calculate the sample size?
The contents of the education materials and intervention need to be explained more. How did you use the theories in designing the intervention? What were the dimensions of the theories?
How did you measure/score the knowledge? Do you have any score for correct/incorrect answers?
Please use the same abbreviation for intervention group through the manuscript: IG or EG
Please correct the footer of table 3. Also in table 3, what “Var after” and “Var C” stand for? Do the p values represent the significance in differences between two groups (also it is obvious in the title of the table; please clarify it in the footer)?
Was there any difference between the level of knowledge in males and females?
Line 265: what is biological?
Do the sample size of your study or quasi-experimental design limited the results interpretation?
Were there any studies which demonstrate the nutritional knowledge of the older population in Ecuador?
The discussion needs to rewrite since the justification about the results is limited.
Author Response
Answer 1. Indeed, the study only shows the effect of the educational intervention on the nutritional knowledge of older adults in Ecuador and the effect of the intervention on their practice was not measured.
Answer 2. The abbreviation of old adults to OA was corrected in line 29.
Answer 3. The situation of nutrition in the elderly in Ecuador and the importance of carrying out this intervention, was included in the third paragraph of the introduction of the corrected article.
Answer 4. Details about the sample calculation were included in the methodology.
Answer 5. The contents of the educational materials and the intervention have been explained in more detail in 2.3 Intervention. The use of theories to design the intervention and the dimensions of the theories have also been explained.
Answer 6. The knowledge assessment criteria and scores for correct and incorrect answers were detailed in 2.3 Intervention of the corrected article.
Answer 7.
Comment: The use of the abbreviation IG, instead of EG, was corrected in the document.
Answer 8. At the end of the tables the meanings of the abbreviations are described.
Answer 9. The analysis of the tables describes the differences in terms of gender
Answer 10.
Comment: It has been included in the correction that the biological refers to the indicators expressed in anthropometric measurements and laboratory tests.
Answer 11. The sample calculation was included in the corrected article and its influence on the interpretation of the results is explained in the discussion, as a limitation.
Answer 12. The studies that demonstrate the nutritional knowledge of the elderly population in Ecuador were included in the third paragraph of the introduction of the corrected article.
Answer 13. In the first part of the new version of the discussion, the justification of the results was improved.
Round 2
Reviewer 1 Report
Thankyou for your response, "Due to the large volume of data described in the tables, a graphical representation was not possible."
Advised :
-- The graph may not display the whole data-- it may be like a graphical abstract of each table-- so that readers can easily understand the matter stated in tables. as readers have less time to go through the large tables and find it difficult to get the abstract of the table.
Author Response
See attachment below
